

# The effects of familiarity on escape responses in the Trinidadian guppy (*Poecilia reticulata*)

Hayley L. Wolcott[1], Alfredo F. Ojanguren[1] and Miguel Barbosa[1,2]

[1] Centre for Biological Diversity, School of Biology, University of St. Andrews, St Andrews, Fife, United Kingdom
[2] CESAM, Department of Biology, University of Aveiro, Aveiro, Portugal

## ABSTRACT

Predation is the main cause of mortality during early life stages. The ability to avoid and evade potential threats is, therefore, favoured to evolve during the early stages of life. It is also during these early stages that the process of familiarization occurs. It has long been recognized that associating with familiar individuals confers antipredator benefits. Yet gaps in our knowledge remain about how predator evasion is affected by social experience during early stages. In this study, we test the hypothesis that familiarization acquired during early life stages improves escape responses. Using the guppy *Poecilia reticulata*, we examine the effect of different recent social conditions in the three main components of predator evasion. Using high-speed motion analysis, we compared the number of individuals in each test group that responded to a visual stimulus, their reactive distance and magnitude of their response (maximum speed, maximum acceleration and distance) in groups composed either of familiar or non-familiar individuals. Contrary to the prediction, groups composed of familiar individuals were less responsive than groups of unfamiliar individuals. Reactive distance and magnitude of response were more dependent on individual size rather than on familiarity. Larger individuals reached higher maximum speeds and total distances in their escape response. Our result indicates that familiarity is likely to affect behaviour earlier in a predator-prey interaction, which then affects the behavioural component of the response. Taken together, our study contributes to previous ones by distinguishing which components of an escape response are modulated by familiarity.

## INTRODUCTION

Predation is a powerful agent of mortality, particularly during early life stages when organisms are at heightened risk due to their smaller size (*Cushing, 1974*). Natural selection is therefore expected to favour the development of antipredator behaviours early in life (*Braithwaite & Salvanes, 2005*; *Vilhunen & Hirvonen, 2003*). Antipredator behaviours are generally divided into two major types: (1) avoidance and (2) evasion (*Fuiman & Magurran, 1994*; *Weihs & Webb, 1984*). Avoidance includes any pre-emptive behaviours in which the individual reduces the likelihood of encountering a predator and consequently of its attack

Corresponding author
Miguel Barbosa,
mb334@st-andrews.ac.uk

(*Fuiman & Magurran, 1994*). Evasion, on the other hand, occurs once the predator initiates the attack. As predator avoidance is not always possible, successful predator evasion tactics are essential for survival. The behaviour and frequency at which each evasion tactic is employed is context-dependent; individuals adopt behaviours that improve their evasive response and, thus, enhance survival (*Domenici, 2010*).

One way in which organisms may reduce the risk of predation is by associating with others, either by schooling or just by joining a group (*Ruxton & Johnsen, 2016*; *Ward & Webster, 2016*). Though groups might be more conspicuous to a predator, each individual within the group has a smaller probability of being predated than if alone. Among the group antipredator benefits of enhanced vigilance, dilution of risk, predator confusion and coordinated antipredator maneuverers (*Krause & Ruxton, 2002*; *Ward & Webster, 2016*), there is strong evidence showing that familiarity within the group enhances antipredator behaviours (*Griffiths et al., 2004*). Familiarity between conspecifics can be broadly defined as the ability to discriminate between individuals based on previous interactions (*Griffiths, 2003*). The process of familiarization is based on visual, and auditory and olfactory cues (*Coffin, Watters & Mateo, 2011*; *Reby et al., 2001*; *Zajitschek & Brooks, 2008*). Repeated exposure to a stimulus can lead to familiarisation, in a social context that may be conspecifics with whom an individual interacts, such as during foraging. Fitness benefits of joining a group composed of familiar conspecifics over unfamiliar individuals has been demonstrated in various taxa (*Figueroa et al., 2013*; *Grabowska-Zhang, Sheldon & Hinde, 2012*; *Grabowska-Zhang, Wilkin & Sheldon, 2011*; *Strodl & Schausberger, 2012*; *Strodl & Schausberger, 2013*), particularly in shoaling fish (*Barber & Wright, 2001*; *Griffiths & Magurran, 1997b*).

The benefits in associating with familiar individuals for the social learning and for the development and acquisition of successful antipredator responses in shoaling fish are acknowledged (*Swaney et al., 2001*; *Ward & Hart, 2003*). Groups composed by familiar individuals may be more cohesive and have reduced neighbour distance (*Chivers, Brown & Smith, 1995*; *Höjesjö et al., 1998*), characteristics which enhance predator confusion and dilute individual risk. Further, familiar groups generally experience reduced within-group aggression and evolve more stable social hierarchies (*Griffiths et al., 2004*; *Höjesjö et al., 1998*; *Johnsson, 1997*; *Tanner & Keller, 2012*). Reduced aggression within familiar groups allows more time for predator vigilance, which may improve escape latency (*Griffiths et al., 2004*; *Strodl & Schausberger, 2012*). Additionally, individuals are more likely to perform cooperative antipredator behaviours when in familiar groups, as they may remember whether the others have behaved cooperatively in the past (*Dugatkin & Alfieri, 1991*). For example, individuals in familiar groups may be more likely to perform more risky antipredator manoeuvres (*Chivers, Brown & Smith, 1995*), join predator mobbing (*Grabowska-Zhang, Sheldon & Hinde, 2012*), or perform predator inspection (*Dugatkin & Godin, 1992*). Such antipredator behaviours put individuals at higher risk, but improve group antipredator response.

While the effect and importance of familiarity on predator avoidance is well recognised, how familiarity shapes predator evasion, particularly the escape response, remains largely unexplored For example, studies to date have focused exclusively on the effect of familiarity

on the latency of the response (*Griffiths et al., 2004*; *Strodl & Schausberger, 2012*) and have not considered other aspects of the escape performance. Successful escape responses depend on various components, such as latency, velocity and distance travelled in the response (*Domenici & Blake, 1997*). For instance, latency, considered as the time between the onset of the predator attack and the start if the response, is crucial for the outcome of the interaction (*Fuiman et al., 2006*). Also, an effective response requires moving away from the attack trajectory fast enough so the predator cannot adjust it (*Fuiman & Cowan, 2003*). Studies on escape behavior have focused on the aspects of the escape response which are modulated by the relative cost of escaping and perceived risk, such as latency, reactive distance (the distance between the predator and prey when the prey initiates a response) and responsiveness (whether or not a prey responds to an attack) (*Domenici, 2010*). Kinematic aspects of escape responses, on the other hand, are less often considered, as they have been considered to be constrained by the sensory-motor system of the individual (*Domenici & Blake, 1997*). A review by *Domenici (2010)* emphasizes that performance in escape responses is not always maximized to the physical capabilities of the individual, which suggests that other factors may cause variability in escape responses. Given the importance of social behaviour in reactive distance and responsiveness (*Dial, Reznick & Brainerd, 2016*), it is plausible that the kinematic aspects of an escape response may be modulated in a similar way by familiarity. In order to fully assess the escape performance of fish, we need to employ an approach that takes into account the multiple behavioural aspects on an escape response. The aim of this study was to address the role of familiarity acquired during early life stages in affecting the different components of the antipredator escape responses in the Trinidadian guppy (*Poecilia reticulata*).

Guppies shoal immediately after birth (*Magurran et al., 1994*). These early stages are important for the establishment and reinforcement of individual discrimination and familiarity in guppies (*Barbosa, Camacho-Cervantes & Ojanguren, 2016*; *Barbosa, Ojanguren & Magurran, 2013*; *Chapman et al., 2008*; *Chapman, Ward & Krause, 2008*; *Laland, Brown & Krause, 2003*). Within group familiarity is likely to affect how a group of individuals respond to a potential predator. Guppies respond to a predator attack by performing a "fast-start" escape response, characteristic to most fish species (*Dial, Reznick & Brainerd, 2016*). This evasion tactic consists of an unambiguous quick and sudden burst of swimming activity usually of only tenths of a second that propels the fish away from an oncoming predator (*Domenici & Blake, 1997*; *Fuiman, Meekan & McCormick, 2010*; *Webb, 1978*; *Weihs, 1973*). Fast-start escape responses integrate a combination of behavioural and kinematic components (*Marras et al., 2011*), both of which were examined in this study.

In view of the antipredator benefits of familiarity, we predicted that juvenile guppies are also more responsive and perform more successful escape responses when in groups of familiar conspecifics. To test this prediction, we exposed familiar and unfamiliar groups of juvenile guppies to a digital display of a looming object and quantified the difference in responsiveness (number of fish responding), reactive distance (based on the size of the stimulus when the response started) and magnitude of the escape response (maximum speed and acceleration achieved during the response, and distance covered by the escaping fish). This approach allows us to identify the role of familiarity in a behaviour closely

related to survival during early life stages and to pinpoint which components of an escape response are more likely to be affected by social experience.

## METHODS

All guppies used were 8th generation descendants of individuals collected from the Lower sections of the Tacarigua River in Trinidad. Several species of fish predators have been reported in this locality including the pike cichlid (*Crenicichla alta*), the blue acara (*Aequidens pulcher*) and the wolf fish (*Hoplias malabaricus*), which also prey intensively on juvenile guppies (*Magurran et al., 1994*). Experimental fish were housed, and all observations recorded, at the aquarium facility at the Sir Harold Mitchell Building, University of St Andrews, UK. The aquarium has an air temperature control system, which kept the tank temperatures at a mean ($\pm$SD) temperature of 24.5 °C ($\pm$ 0.3 °C). All stock tanks contained similar numbers of males, females and juveniles. Lighting conditions followed a 12-hour light/dark cycle. All fish were fed daily with TetraMin® flake food. Our experimental design was examined by the Biology School Ethics Committee from the University of St Andrews and declared our study to be exempted of Animal Ethics approval.

### Test fish collection and rearing

Prior to the experiment, we collected three juveniles from three different stock tanks (60 $\times$ 40 $\times$ 40 cm) that contained a mix of males, females and juveniles using a dip net (there are 15 Lower Tacarigua stock tanks in at the University of St Andrews aquarium facility). This ensured that the test groups were composed neither of familiar conspecifics nor of close kin. Further, in all stock tanks there are large and smaller boulders and java moss, which allows a more natural environment for guppies. Immature juvenile guppies (i.e., age between five and six weeks) were allocated to a holding tank (20 $\times$ 22 $\times$ 30 cm) a to create a test group. Each test group was composed of three individuals. A total of 42 holding tanks were used. Black plastic sheets were placed between each tank to ensure each test group was visually isolated from adjacent groups. Fish were of similar size and randomly distributed between holding tanks (mean ($\pm$SD) 10.8 ($\pm$ 1.7) mm). Nevertheless, in order to be able to identify each individual during tracking, test groups were carefully constituted of different sized individuals. Each test group remained in its holding tank for two weeks to ensure the establishment of familiarity between tank mates (*Griffiths & Magurran, 1997a*).

### Escape response trials

We split the juveniles into two treatments: a familiar and an unfamiliar. Each group was composed of three juveniles (a total 42 groups, 21 familiar and 21 unfamiliar). Each day we tested six groups, three groups with familiar individuals and three of unfamiliar individuals. In familiar groups, individuals were tested with those fish that they shared the holding tank with for two weeks prior to testing. For unfamiliar groups, we took three fish, each from a different holding tank so they had not seen each other before, and put them together in the observation chamber for testing (Fig. 1). Unfamiliar groups were treated as a control. Each group was only tested once.
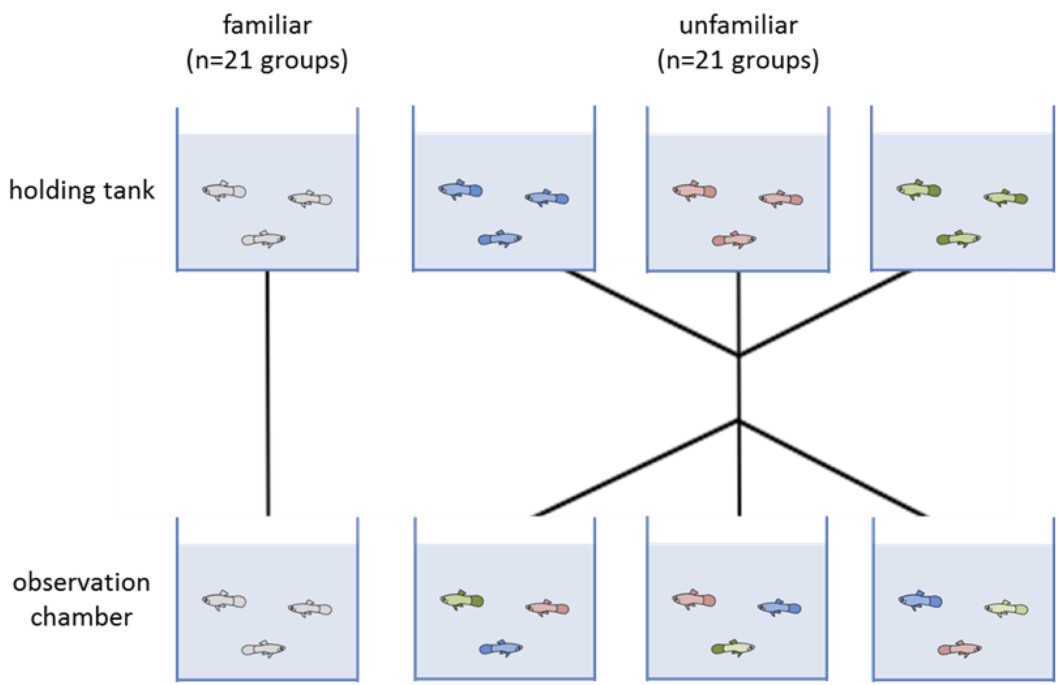

**Figure 1** **Diagram of the two experimental treatments (familiar and unfamiliar).** Individuals were allocated to a holding tank with two other conspecifics for two weeks. Each testing day, three groups were tested where fish remained with those they had been sharing a tank with (familiar treatment). The other three groups had the individuals swapped so that none of the fish had encountered each other previously (unfamiliar treatment). Forty-two groups were tested in total, 21 of each treatment.

All tests occurred between 9:00 and 11:00 am and at least an hour after being fed. These measures were taken to avoid differences in satiation rate and time of day that might affect the behaviour of the individuals. The experimental setup used to assess escape response was based on an established protocol (*Fuiman, Meekan & McCormick, 2010*), but modified for this experiment (Fig. 2). Each trial involved presenting a digital display of a looming object to a test group. The digital display consists 1.8-second sequence showing black oval in the middle of a white background that increases its size to simulate an approaching object (Supplemental Information). The same stimulus has been shown to elicit a startle response in larval fish of similar size (*Fuiman et al., 2006*; *Ojanguren & Fuiman, 2010*). The video was presented using a LCD screen (Braun 1210) located 0.23 cm from a $10 \times 10 \times 10$ cm glass test chamber. Water depth within the observation chamber was kept at 225 ml to minimise vertical movement in escape responses. For each trial, a test group was transported to the observation chamber one individual at a time and given at least 10 min of acclimatisation to their new surroundings before testing began. Each individual fish was only tested once. After the terminus of the trials the individuals were returned to a stock tank and were not reused in the experiment.

Individual response to the visual stimulus was recorded at 240 frames s$^{-1}$ using a high-speed video camera (Casio EX-FH25 EXILM) through a 45°-angled mirror to obtain an overhead view of the observation chamber. The observation chamber sat on top of a black
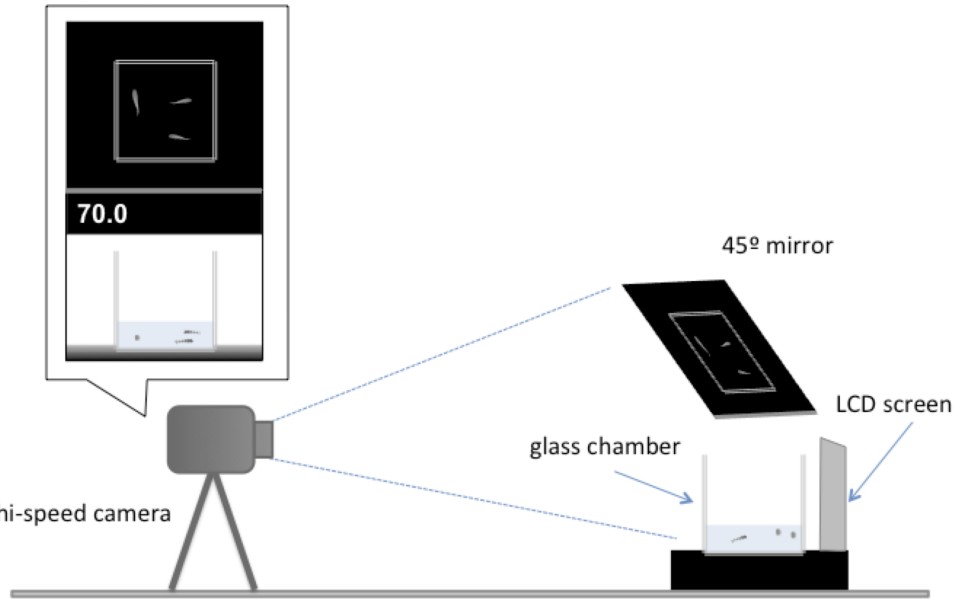

**Figure 2   Illustration of the experimental setup.** A camera was placed 1 m away from a glass tank (10 ×10 ×10 cm) positioned before the LDC screen that showed the digital display of a looming object. The front of the tank and the overhead view of the tank were recorded in high-speed video for each trial. The distance in centimetres of the digital looming object was displayed on the top left of the screen.

surface and was illuminated by lamps positioned left and right of the chamber so that the response could be clearly observed. All individuals tested were transferred to a small petri dish with a small amount of water and photographed from above. Individual standard length was measured to nearest millimetre using ImageJ analysis software (*Abràmofff, Magalhães & Ram, 2004*). All tested individuals resumed normal routine swimming activity immediately after the escape responses. No fish died during the tests, or after the picture was taken. After the terminus of the study, all individuals were returned to stock tanks.

## Data analysis

Video recordings were analysed frame by frame to determine responsiveness (the number of fish that responded to the stimulus in each test group) and the reactive distance (the virtual distance between the looming object and the first individual that responded, calculated from the size of the oval on the screen at the moment of the start of the response and the distance of the fish from the screen) (see *Fuiman, Meekan & McCormick, 2010* for details). The position of the fish in 2-dimensional coordinates for the overhead view was obtained using the manual tracking plugin in ImageJ (*Cordelières, 2005*), this allowed us to calculate maximum speed, maximum acceleration and total distance covered in the response (see *Fuiman, Meekan & McCormick, 2010*; *Fuiman et al., 2006*).

## Statistical analysis

Differences in responsiveness between familiar and unfamiliar groups were tested with a Wilcoxon rank sum test to account for the fact that responsiveness was a discrete variable.

The responsiveness of each test group was ranked according to the number of individuals within the group that responded (either 0, 1, 2 or 3). We considered that the response was over when the distance travelled between three consecutive frames (12.5 ms) was 1 mm or less.

In order to investigate the effect of familiarity on reactive distance and in the magnitude of the response (maximum speed, maximum acceleration and distance covered in a response) we used Generalised Linear Fixed Effect Models (GLM). Reactive distance, maximum speed, maximum acceleration and distance travelled during a response were only measured on the first fish that responded. On the only trial that two fish responded in the same frame, the fish that had the larger reactive distance was considered the first responder. Each group was only tested once. Each full model included familiarity as main effect treatment and standard length as a covariate (i.e., of the individual that first responded), as well as their interaction. The linear predictor and expected values scales were linked using a log function. Diagnostic plots revealed significant departures from normality of the residuals for both response variables reactive distance and total distance. Normality and homogeneity assumptions about the distributions of residual values on the dependent variable were improved by log-transforming the response variables. To account for the effect of size in escape responses, all models included individual standard length as covariate. All analyses were performed in using R (*Team, 2016*).

## RESULTS

Individual standard length between familiar and unfamiliar treatments did not differ (mean ($\pm$ SD), Familiar = 117.3 (19.7); Unfamiliar = 122.2 (27.9), $p = 0.089$).

### Responsiveness

A total of 42 groups composed of three different sized individuals were tested. Of the 30 groups in which one or more individuals responded, 19 groups were familiar and 16 groups were unfamiliar. There was a significant effect of familiarity on responsiveness (Wilcoxon rank sum: $W = 451.5$, $p < 0.001$) (Fig. 3), where responsiveness was higher in unfamiliar groups. In the majority of familiar groups only one individual in the group responded, whereas the unfamiliar groups showed more instances where two or more individuals reacted to the stimulus.

### Reactive distance

We failed to detect an effect of familiarity and individual standard length on reactive distance (Table 1, Fig. 4A).

### Magnitude of the response

We failed to detect an effect of social treatment on maximum speed ($p = 0.263$), maximum acceleration ($p = 0.699$) and total distance ($p = 0.698$) (Table 1, Fig. 4). For maximum acceleration the effect of individual standard length was similar between treatments ($p = 0.078$). There was an increased in both maximum speed and total distance travelled as a function on increased standard length (Figs. 4C, 4D, Table 1). Notably, we observed

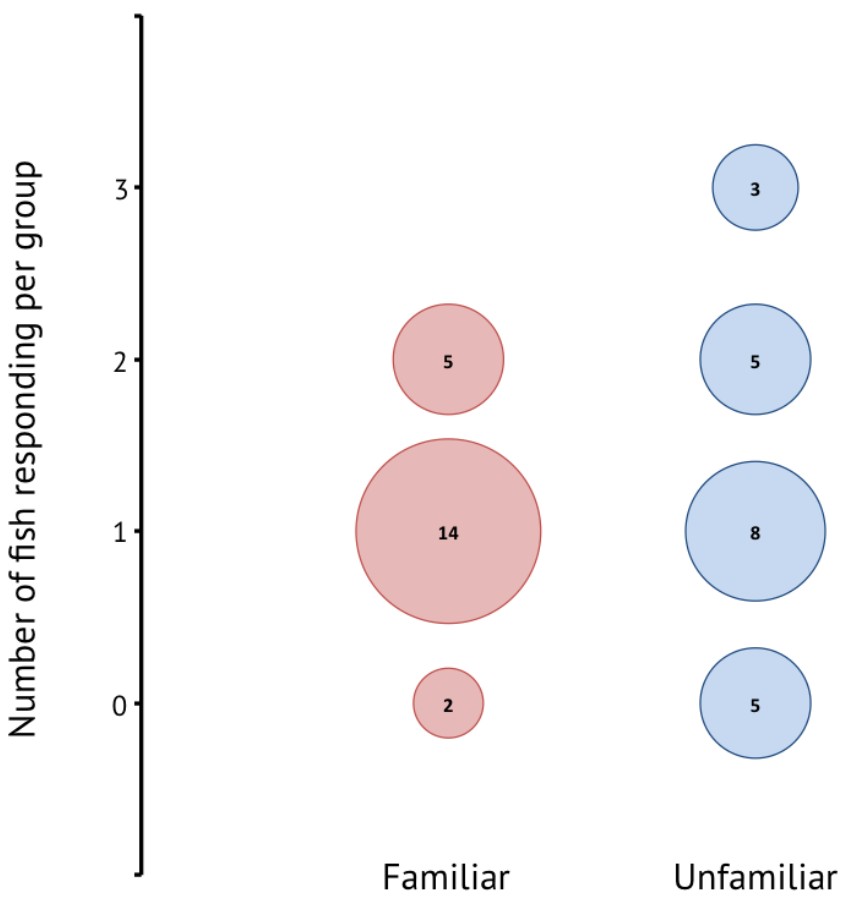

**Figure 3  Responsiveness for familiar and unfamiliar groups in terms of how many individuals in a group of three responded to the stimulus.** The numbers within the bubbles give the number of groups.

an almost twice-greater gradient in the familiar treatment than in the unfamiliar treatment in terms of maximum speed. There was an increase of 82.05 ms in maximum speed per millimetre of standard size in the familiar treatment, while in the unfamiliar treatment the gradient was of 44.14 ms per millimetre of standard size (Table 1, Fig. 4B).

## DISCUSSION

A novel contribution of this study is that it examines the consequences of familiarity during early stages in the performance of escape responses separating the multiple aspects of the response to determine which parts depend on the social environment. Through high-speed analysis of the escape responses in familiar and unfamiliar groups of guppies, we demonstrate that early social experience plays a role in shaping how groups of fish respond to a stimulus. Namely, we showed that unfamiliar groups had more individuals perform an escape response than those in familiar groups. Unexpectedly, other components of the escape response, namely latency and magnitude, were not affected by familiarity. Furthermore, the maximum speed and distance covered in the response were correlated with individual size rather than with level of familiarity within the group. In combination,

**Table 1** Generalised linear models for testing the effect of familiarity on different qualitative measures of response.

| Response variable | Explanatory variable | Estimate | Std Error Sq | *T*-value | *p*-value |
|---|---|---|---|---|---|
| Reactive distance | Intercept | 2.526 | 1.819 | 1.389 | 0.175 |
| | Unfamiliar | 1.985 | 2.879 | 0.690 | 0.496 |
| | Length | 0.069 | 0.142 | 0.488 | 0.629 |
| | Unfamiliar: length | −0.178 | 0.243 | −0.737 | 0.467 |
| Maximum speed | Intercept | 175.7 | 269.9 | 0.651 | 0.519 |
| | Familiar | −487.8 | 427.2 | −1.140 | 0.263 |
| | Length | 44.14 | 21.16 | 2.086 | 0.045 |
| | Familiar: length | 37.91 | 36.06 | 1.051 | 0.301 |
| Maximum acceleration | Intercept | −147.6 | 4,846 | −0.003 | 0.997 |
| | Familiar | 2,989 | 7,671 | 0.390 | 0.699 |
| | Length | 6,909 | 3,799 | 1.819 | 0.078 |
| | Familiar: length | −3032 | 6,473 | −0.468 | 0.642 |
| Total distance | Intercept | 0.699 | 0.908 | 0.771 | 0.447 |
| | Familiar | 0.562 | 1.437 | 0.391 | 0.698 |
| | Length | 0.194 | 0.071 | 2.733 | 0.010 |
| | Familiar: length | −0.055 | 0.121 | −0.454 | 0.653 |

our study suggests, that familiarity plays a less meaningful role in determining some behavioural components of the escape response.

Our results show that familiarity affects group responsiveness. There were a greater number of individuals responding within each group among unfamiliar groups than among familiar groups (Supplemental Information). While most fish species rely on the escape response to avoid a potential predator (*Domenici, 2010*; *Fuiman & Magurran, 1994*), escape responses may however vary within and among individuals (*Lima & Dill, 1990*; *Ward & Webster, 2016*; *Ydenberg & Dill, 1986*). If there is enough information to accurately predict the level of threat in a given environment, then it is advantageous for a prey to only flee when it is necessary for survival avoiding false alarms that could in turn attract the attention of nearby predators (*Ward et al., 2011*). For example, minnows performed antipredator behaviours in response to a realistic pike model, whereas an unrealistic stimulus elicited no response (*Magurran & Girling, 1986*). The lower responsiveness in familiar groups may be a result of improved vigilance. According to the theory of limited attention, performance is reduced when attention must be divided among different tasks (*Dukas, 2002*). Therefore, if individuals are not spending time inspecting or acting aggressively toward group mates, as is often found among unfamiliar individuals (*Griffiths et al., 2004*; *Johnsson, 1997*; *Tanner & Keller, 2012*), then they are likely to have more time to dedicate to other tasks, such as predator vigilance (*Strodl & Schausberger, 2012*; *Strodl & Schausberger, 2013*; *Zach et al., 2012*). Guppies from familiar groups may have been able to accurately assess the non-threatening nature of the stimulus. An alternative explanation is that fish in familiar groups feel safer as they are with individuals they have seen before and this may be why familiar individuals are more likely to perceive the oval shape stimulus as non-threatening. On the other hand, unfamiliar groups may have been more skittish and

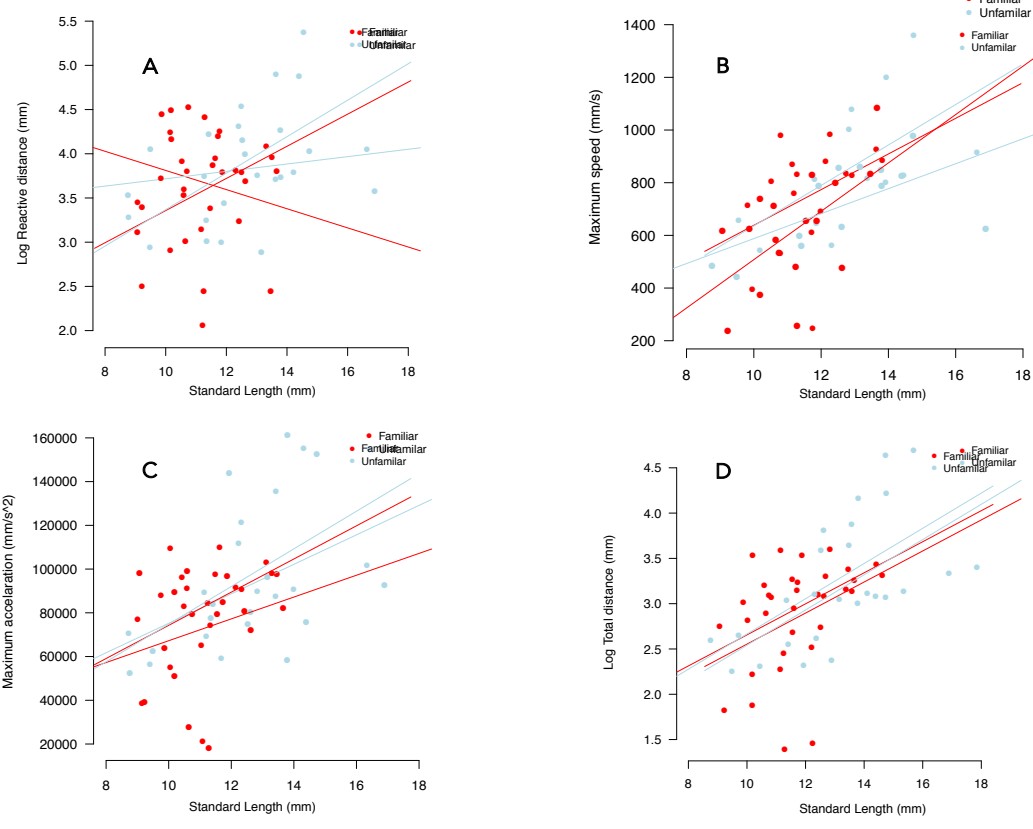

**Figure 4** **Variation in reactive distance (A), maximum speed (B), maximum acceleration (C) and total distance (D), in familiar (red circles) and unfamiliar (blue circles) groups.** Lines were fitted using the co-efficients of linear models.

thus more likely to be startled by the stimulus. Interacting with unfamiliar individuals can be stressful (*Choleris et al., 1998*), particularly if such interactions are associated with increased aggression (*Galef, Kennett & Wigmore, 1984*). Individuals may perceive higher risk when shoaling with unfamiliar conspecifics, as was found in fathead minnows who had a higher production of epidermal alarm substance cells when in unfamiliar shoals than familiar shoals (*Wisenden & Smith, 1998*). Furthermore, escape responses from the digital display may be misinterpreted as an attack by the other group mates. Aggression is common among guppies, in both natural as well as laboratory conditions (*Magurran, 2005*; *Thibault, 1974*). Therefore, it is plausible that an individual guppy would flee from an unfamiliar group mate that is performing a fast-start response, as this could be misinterpreted as an attack.

We failed to detect an effect of familiarity on the reactive distance of an escape response. Comparable studies have found that familiarity reduces the latency of an escape response. Similarly, familiar juvenile brown trout responded 14% faster than unfamiliar ones when exposed to a simulated predator attack (*Griffiths et al., 2004*). In both their and our study, reduction in reaction time has been attributed to the associated benefits of the theory

of limited attention. Our results, therefore, suggest that familiarity is more important in antipredator behaviours earlier in a predator sequence. A predator must successfully encounter, attack and capture a prey, where a prey's strategy is to interrupt this sequence. It has been suggested that avoiding the encounter and attack, are prey's best strategy (*Fuiman & Magurran, 1994*). Previous experiments included an entire predator interaction, such as a model heron swinging forward and plunging its beak into the water (*Griffiths et al., 2004*) or a live predator (*Strodl & Schausberger, 2012*), and could, therefore elicit such behaviours. In contrast, our experiment only elicited behaviours seen in the last few milliseconds of the attack.

Familiarity has been found to enhance avoidance tactics. For example, predator confusion was enhanced in shoals of familiar fathead minnows that had reduced neighbour distance and more shoal cohesion in response to predator stimuli compared to unfamiliar shoals (*Chivers, Brown & Smith, 1995*). Tighter shoal cohesion reduces the probability of being captured by a predator (*Mathis & Smith, 1993*). In addition, familiar shoals exhibited a greater number of predator inspections with more inspectors per inspection when faced with a model pike (*Chivers, Brown & Smith, 1995*). Predator inspection, where an individual or small group of individuals approach a predator, pause and swim away (*Pitcher, 1991*), enables prey to gain valuable information on the threat of a predator. This behaviour, though risky to inspectors, is associated with improved avoidance of a predator attack (*Godin & Davis, 1995*; *Magurran, 1990*; *Magurran & Pitcher, 1987*). Therefore, it is likely that familiarity is more crucial in antipredator behaviour associated with predator avoidance than predator evasion.

The effect of familiarity on the magnitude of the response was not significant. The kinematic aspects of escape responses are often assumed to be constrained by the sensory-motor system of the individual (*Domenici & Blake, 1997*). However, juvenile guppies reared in an environment with intense social aggression travelled a greater distance in the first five frames after a simulated avian attack than those reared in absence of social aggression (*Chapman et al., 2008*). It is then recognized the need to implement an integrative approach that accounts for all aspects of an escape response in order to obtain a clear understanding of the mechanisms of response to a predator (*Domenici, 2010*). While other behavioural variables may affect the magnitude of an escape response, our study provides evidence that familiarity is not one of them. Our results showed that size, rather than familiarity, influenced the magnitude of the response than familiarity. This result is consistent with previous studies that have shown that the magnitude of the fast-start response in young fish increases with body length (*Dial, Reznick & Brainerd, 2016*). While behavioural effects on the locomotive performance cannot be ruled out (*Domenici, 2010*), our study and others (*Gibb et al., 2006*; *Ojanguren & Braña, 2003*) lend strong support that the magnitude of a fast-start response is largely determined by morphology, rather than by social conditions.

In this study, we provided a test for the relative effect of familiarity in modulating predator avoidance behaviour by measuring several components of the escape responses using high speed video analysis. Our results suggest that familiar groups may have improved antipredator performance, as individuals conserve energy and are less conspicuous by not fleeing in a non-threatening situation. Nevertheless, further studies are necessary to

elucidate this. Future studies may try to tease the contribution of group size and familiarity in modulating the predator escape response, by testing familiar and non-familiar groups composed of different number of individuals. Our results also suggest that the effects of familiarity on the response are perhaps unlikely to play a role on escape performance in the last few milliseconds of a predator attack. Instead, we believe that familiarity is more likely to affect behaviour earlier in a predator–prey interaction, which then affects the quality of the response.

## ACKNOWLEDGEMENTS

We are grateful to Maria Dornelas, Anne Magurran and Mike Webster for providing helpful comments on early drafts. We are also thankful to Maria Joao Janeiro for helping with the statistics and to the Biodiversity and Behaviour Group for the rewarding discussions and comments throughout this research.

### Funding

This study was funded by a Postdoctoral fellowship to Miguel Barbosa (SFRH/BPD/82259/2011). The funders had no role in study design, data collection and analysis, decision to publish, or preparation of the manuscript.

### Grant Disclosures

The following grant information was disclosed by the authors:
Postdoctoral fellowship to Miguel Barbosa: SFRH/BPD/82259/2011.

### Competing Interests

The authors declare there are no competing interests.

### Author Contributions

- Hayley L. Wolcott conceived and designed the experiments, performed the experiments, analyzed the data, wrote the paper, prepared figures and/or tables.
- Alfredo F. Ojanguren and Miguel Barbosa conceived and designed the experiments, analyzed the data, contributed reagents/materials/analysis tools, wrote the paper, prepared figures and/or tables, reviewed drafts of the paper.

### Animal Ethics

The following information was supplied relating to ethical approvals (i.e., approving body and any reference numbers):

Approval was provided by the University of St. Andrews Animal Welfare and Ethics Committee (2015). The review panel declared no need to obtain Animal Ethics approval.

### Data Availability

The raw data supporting this publication can be accessed at
http://dx.doi.org/10.17630/92831d81-38f0-4573-b2e5-e1d11adf9322.

## Supplemental Information

Supplemental information for this article can be found online at http://dx.doi.org/10.7717/peerj.3899#supplemental-information.

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
