# Peer review of "The effects of familiarity on escape responses in the Trinidadian guppy (Poecilia reticulata)"

_PeerJ, doi:10.7717/peerj.3899_

## Round 0.1 · original submission · Major Revisions

· Academic Editor

Major Revisions

Three reviewers have now had the chance to provide their feedback on your paper. Both reviewer 1 and 2 request that you provide greater consideration to the size of the fish in each group in relation to their responses. Additionally, reviewers 2 and 3, request that you clarify and run alternative analyses to tease apart the observed effects in more detail. More specifically, all three reviewers provide very clear and detailed comments that will help you to improve and strengthen your manuscript. I have no specific comments to add to theirs at this stage, however I think it is worth highlighting that all reviewers noted the merit of your paper and I agree.

·

Basic reporting

All good

Experimental design

All good, but see suggestions in general comments

Validity of the findings

All good, but see suggestions in general comments

Additional comments

Overall this is a good study. My only issue is that the expectations are a little naive. There is every reason to suspect that the magnitude of response is largely due to inherent properties of the fish (size, size of the mauthner neuron etc) thus I fail to see any reason why the authors would expect to see familiarity effects on these traits. It makes sense for other parts of the avoidance strategy (eg time to first detection, reaction distance etc) as these are modified by social interactions. With this in mind I would alter the introduction a little.
Secondly it is hardly surprising that size swamps all other aspects of the escape mechanics. I would use some kind of residual to remove size effects completely. It’s unlikely you’ll find anything, but worth a try. Be sure to show us that familiar and unfamiliar groups do not differ in size.
*Note I have marked the pdf for editorial suggestions

Specific comments:
L63: familiarity has other benefits as well, eg enhanced social learning (swaney et al 2001).
L108: I can understand the reasoning behind the idea that familiar groups might respond in greater synchrony, perhaps even respond more quickly at longer distance, but I cant imagine how it would effect things like the magnitude of the response. This is largely innate and controlled by the mauthner neuron.
You might even argue that familiar fish feel safer in their shoals and therefore respond later. Hard to say.
L115: You need to tell us how long ago. Anti-predator behaviour declines rapidly in captivity.
L143: what did you do with the fish afterwards? Did you put them back in the home tanks? Reuse them? Was each fish only tested once? We need these details here.
L216: when it comes down to it the Ns are pretty low
L231: where length is a covariate it will be important to show that familiar and unfamiliar groups did not differ in length. Note you really need to remove length. Perhaps use residuals.
L247: If you ask me this is not unexpected at all. See reasons stated above.
L273: The authors might like to read Brown 2002 (Journal of Ethology 20, 89-94)

Reviewer 2 ·

Basic reporting

In investigating the role of familiarity in influencing group and individual level anti-predator responses the authors seek to understand what details of responses may be influenced by group composition. Overall the manuscript is clearly written and their justification for their study is well supported. However there are two areas in particular where rewording could help better reflect the study they conducted and results they found.

1) In the abstract the authors state that “Using the Trinidadian guppy we examine the effect of different early social conditions in the three main components of predator evasion”, implying that they manipulate early life experience. Instead the paper manipulates recent social experience (i.e. the two weeks prior to the experiments; lines 136-137), which is a different line of study. I would therefore suggest the authors revisit all mention of “early social experience”.

2) In the discussion, the authors state quite strongly that they “unambiguously demonstrated that familiarity plays a significant role in shaping how groups of fish respond to a stimulus”. Since familiarity only predicted the number of individuals that responded, but did not predict several of the other variables they analyzed, this tone appears unwarranted. The statement they include later on in the conclusion, namely that “In combination, our study suggests that, while familiarity affects how groups respond to a visual stimulus, it plays a less meaningful role in determining the quality of the escape response” is a much more measured and appropriate summation of the study. I would suggest that they revisit the tone in the discussion, especially since PeerJ does not require results to be “flashy”, and thus there is no reason for overstating their case.

Finally, and on a more minor note, the (Strodl & Schausberger 2012) paper that they cite does not investigate responses to predators, and I think it may have been cited in error (e.g. line 284, 293). The authors of the paper in question wrote other papers that year which the authors of this ms may have instead intended to cite. I would suggest they double check all references they list.

Experimental design

While most of the experimental design was easy to understand (Figure 1 helped especially), there are a few minor areas where more detail would be helpful.

Line 128-129: Without knowing the origin of the stock in the different tanks, it is hard to assess whether the individuals were related. A little more detail here, such as how long the stock had been separated and from where it was originally sourced would be useful.

Additionally, it would be nice to know if the source and experimental tanks were featureless or did individuals have access to cover? (which could influence anti-predator behavior)

Finally, were the size differences between individuals (one big, one medium, one small) also maintained in the unfamiliar trials?

Validity of the findings

Since the authors only looked at the details of the anti-predator behavior in the first responding fish of any group (as stated lines 189-190), they were potentially sampling a non-random individual from the group. Given that the size of the individual, as opposed to the familiarity of the group, appeared to predict much of their response behavior, it would be very informative to know whether the size of the first responding individual was similar across the familiar and unfamiliar groups. I would suggest that the authors consider this addition to their analysis because it could mask the effects of familiarity on anti-predator behavior. If there are differences between the groups in the size of the first responding individual, then the authors need to consider analyzing the behavior of other group members.

Additional comments

(see comments in previous three sections)

Reviewer 3 ·

Basic reporting

The manuscript is clearly written, but the overall presentation can be improved in quite a few places, either by rewording or shortening of the prose. Below, I've made specific suggestions to the author. The background and context of the study, as laid out in the introduction, are strong and well-supported by the literature cited. Structure is fine and overall the results speak to the hypothesis (predictions). I believe the authors have generally met the basic reporting criteria.

Experimental design

The design is good, but there are analysis issues that need to be addressed. Specifics outlined in comments to author

Validity of the findings

no comment

Additional comments

Analysis. I would strongly suggest choosing to use either a strict hypothesis-testing approach or a modeling approach, but not both. My inclination would be the former, because you have set your study up as a test of a hypothesis (predictions). Also, you have relatively few parameters and a relatively small sample size (adequate, but small for this kind of analysis).

If you choose the AIC approach, I would recommend calculating a AICc for small sample size. I would also provide the variance inflation factor, c, calculated as X2/df, based on the full model, not the reduced models. Check Burnham and Anderson.

Make clear on line 201 that you are using a generalized linear model framework.. Yours is a fixed effects model, so although obvious, spell it out for the reader. Also, need to include the link function used, which I assume was a normal log link, but again, need to make these particulars clear.

I believe you may have a mistake in calculating K. If your analysis (generalized linear models) used the log likelihood method of model selection, which I’m guessing it did, then K = no of explanatory variables, + intercept + variance.

Finally, with the AIC approach you would not be reporting p-values, except possibly for the Wilcoxon test (line 198).

Table 1. treatment + length should be “treatment x length” since it is an interaction. The order of model effects (factors) column seems unconventional to me. I would check this against other published tables. Possibly start with the full model that contains all of the factors, including the interaction, then all remaining combinations thereof, putting them in increasing order of the ΔAIC. You could choose to report just those with an AIC < 4.0. That would be one way to do it.

Results/Discussion. You might consider the effect of group size in your discussion. It would have been useful, in hind sight, to test for this, because group size could interact with familiarity. Contrary to your predictions, groups composed of familiar individuals were less responsive than groups of unfamiliar individuals, suggesting that groups size may be an important factor. Group size could combine with safety in numbers, so two curves of response plotted against group size may start the same (n = 1), separate around 3 or 4 ( your result) but then converge at higher groups, like around 8 with very low responses. Do you think this is possible? If so, perhaps work it into the abstract in place of 31-32?

Additional comments
Abstract.
19 please! driver = cause
19-20: Suggestion: Traits that contribute to the avoidance/amelioration of potential threats should be under strong selection during these stages, along with the process of familiarization.
22: “Less, however, is known” suggest “Yet, gaps in our knowledge remain”.
27: suggest: …we compared the number of individuals in each test group that responded to an artificial stimulus, their reactive distance and magnitude of their response (acceleration and distance) in groups composed of…”
30-32. Consider revising, perhaps bringing in the idea of group size, assuming you think it could be playing a role here.
34. suggest: Our experimental approach revealed specific aspects of the escape response that are more likely to be influenced by variable(?) early social conditions.
53 for = to
57 between individuals within a group
64 Ward and Hart (2003) found….
66 Groups composed of…individuals have been found to be more…
73 “will have an idea” = “may remember”
80 take out “fairly”
81 furthermore = for example
91 “…responses. We used a tropical fish, the Trinidad guppy ( ), because its behavioral ecology and predatory regimes are well known (Magurran 2005).” (This is a suggestion, and you might be able to write it better, but I would try to come up with an explanation along these lines)
105 hypothesis = prediction
127 size?
134 I would take out these sorts of personal observations unless you are prepared to provide some data. A critical reader might be skeptical that really observed this.
139-143 It took me a while to figure out that each test group has 3 fish. You should state here in the methods, clearly, so that the reader has it in mind going forward.
154-155. Again, I don’t think you need to say this. You already have a very artificial set up and the reader knows and appreciates that you are not necessarily trying to exactly duplicate nature. That’s not your point.
162. Take out “gently”. You didn’t do it “roughly”, right?
163. Take out “so no …required”
164 standard
166 escape
166 and = or
167 insert “were”
256 “, escaping may not always be the best strategy” As written, this doesn’t make sense to me if the alternative is not to escape and be eaten. How about writing “responses may be highly variable”?
283-284. I suggest shortening the Discussion, and statements like this could easily come out.
296-307: Good. Keep this paragraph!

Overall, Yours is a nice contribution that will be of interest to the many researchers who are studying predator avoidance behavior.. I believe that if you go with the hypothesis testing approach, deleting the AIC material, you will have a successful paper. I would also recommend shortening the Discussion, and possibly even the Introduction. It could be tighten a bit too.

---

## Round 0.2 · Major Revisions

· Academic Editor

Major Revisions

As only one of the three original reviewers who reviewed your first submission was available to review your revised version of your article (reviewer 3), I invited a new reviewer (reviewer 4) to also evaluate your re-submission at this time. While reviewer 3 believes you have successfully addressed their original concerns raised in their first review, reviewer 4 has raised issues with your analyses and reporting, as well as with your framing, that will require your attention. They have provided a detailed review of your paper that will help improve its clarity, and I request that you address their comments and concerns.

In addition to this reviewer's comments, I had a question regarding your analysis presented on line 232. Where you state “where responsiveness was higher in unfamiliar groups” do you mean the number of individuals in a group that were responsive was higher? If so, please clarify your definition here. Furthermore, on line 171 you note “Each individual fish was only tested once”. This suggests a between subjects method was used. If this is the case, why did you use a Wilcoxon rank sum (a repeated measures test) for a between subjects design (as reported on line 232)?

Reviewer 3 ·

Basic reporting

The authors have satisfactorily addressed all of the issues that I raised in my previous review. They agreed with my comments in almost all cases with the exceptions being only suggestions, and in those cases I understand their reasoning.

Their combination of the hypothesis-testing with information theoretic approaches was a mistake in my view and so I was very relieved to see that they had agreed and omitted the latter.

I read the other reviews, which also raised important questions, and again, it appears to me that the authors have addressed them, although it is possible that the reviewers may not agree.

Experimental design

I believe that the article meets the PeerJ standards with respect to experimental design.

Validity of the findings

no additional comments

Reviewer 4 ·

Basic reporting

No further comment

Experimental design

I have no specific comments about the experimental design, which was generally ok.

Validity of the findings

I have a number of important concerns about the validity of the findings and their interpretation. First of all, it is unclear what specifically the authors analysed to compare the responsiveness between familiar and unfamiliar groups and there was no way I was able to replicate their result with the provided data and only non-significant effects were acquired. Secondly, for the other four analyses the authors used a GLM approach and therefore forgot to account for non-independence between the groups and should therefore have used a mixed modelling approach (GLMM). This is likely to considerably change results. Thirdly, the authors refer to significant effects of familiarity while there are none, to multiple significant effects of length where there is only one, and make wrong interpretations on finding a significant effect. From the data and results quite an extent of the discussion seems not appropriate as the effects described seem to be different than interpreted.

Additional comments

Besides my main concerns about the analyses and their interpretation and the other points below, a more important general point is that the study is still lacking is a good explanation of the mechanistic expectations and interpretation of why familiarity would matter in fast startle responses, especially in the introduction. This point should be well thought about and integrated and, together with clarifications to the reasoning for doing this specific study, besides because it has not been done yet, the manuscript will be improved.

Line 24: This sentence remains ambiguous and best to remove “during early life stages “ or make clear also the experiments were conducted in the same life stage.
Lines 30-31: See points below, I do not think the result holds and that therefore there are no significant effects of familiarity on responsiveness in the specific situation tested.
Line 32-35: Not “more” as familiarity had no effect and furthermore size only had an effect on reactive distance!
Lines 60-62: Familiarity itself does not have anything to do with individuals, therefore be careful with defining it here.
Line 64: Please rephrase this sentence. Repeated exposure to a stimulus can lead to familiarisation, in a social context that may thus be conspecifics with whom an individual interacts, such as during foraging.
Line 64-67: This can give greater fitness benefits/this has been shown for a couple species but be careful with making such a general statement as it may be species or context dependent.
Line 70-71: Again, don’t generalise too easily.
Line 73-75: Same.
Line 77-83: This depends strongly on the species and the type of social system, for example much more likely in birds with strong social structures and many repeated interactions than in fish that live in very large fission-fusion populations.
Line 86-87: This sentence seems to show the opposite of what is stated in the foregoing sentence.
Line 104-107: It remains unclear why exactly familiarity would need to be investigated in the context of escape response and thus why the study was conducted. Stating that “how familiarty shapes predator evasion … remains unexplored” (Lines 85-86) and “Dominici calls for… all espects on an escape response” (lines 103-104) do not give clear reasoning why this would be needed.
Line 120-122: But why? What mechanisms are fundamental to your prediction?
Line 163-169: So was there was a difference in the way fish were placed in the observation chamber for testing as fish from familiar groups would be placed there together and fish from unfamiliar groups one at a time? This could have a number of effects and should be clarified.
Lines 163-169: Thus each group received one test trial? This needs to be clarified.
Lines 198-214: These two paragraphs need to be integrated into one, which can be half the length.
Lines 227-238: The authors need to make clear if they run analyses at the individual or group level. If at the individual level they cannot do a GLM as they have to account for non-independence within groups by running a mixed model and group id as a random factor. Furthermore, the way how responsiveness is analysed is not correct (see below). Also, if the analyses were run at the group level it needs to be clarified how body length was calculated per group.
Lines 244-246: Statistically testing this, such as with a Chi-squared test, shows there is no significant difference between the groups in any fish responding or not. This should be integrated here.
Lines 246-248: I don’t understand what the authors analysed here and I cannot replicate their result. It seems the authors want to compare the average nr of responsive individuals per group between the treatment. But running this actual analysis with a Wilcoxson rank sum test yield a non-signifant result. Running this with all groups with zero responders excluded still results in a non-significant result so it is mysterious what the authors actually did. Also, here it can be clarified that for double the number of groups in the unfamiliar groups no individual responded compared to the familiar groups, although that effect was als non-significant.
Line 253-258: This section is confusing and seems to make false interpretations. From Table 1 it is clear that neither treatment or length had a significant effect on the reactive distance nor was there an interaction between them. This needs to be stated first of all. Secondly, that the intercept was significant simply means that it for all individuals on average it was significantly different from zero, which is not that informative for your questions. Thirdly, as you don’t find an interaction to be significant there is no difference in slopes, which is now argued on line 255.
Line 263-265: Length was only significant for maximum speed, thus again, this statement is false.
Line 266-271: The table clearly shows there was no significant interaction between familiarity and body length thus does not support this statement and interpretation!
Results: From the model output in Table 1 the (lack of significant) results are clear to me: familiarity and body length did not interact and did not have an effect on the reactive distance, maximum speed, maximum acceleration or total distance. The only significant effect observed was unsurprisingly that larger fish reached a higher maximum speed.
Line 278-279: The authors actually do not show this as their only significant result for familiarity cannot be replicated. Furthermore, as the previous reviewers also already highlighted the authors should not make too strong statements and try to oversell their results.
Line 280-281: This is purely hypothetical and has no place in the first paragraph of the discussion.
Line 283-284: Again the authors make a statement about an apparent significant result that is not signifant (at least to table 1)!
Line 285-286: Even if this effect would be significant it does not really say anything about how groups respond but how individuals in such a group respond.
Line 287: See other points, the result on which this statement is based needs to be clarified/rectified and may considerably change the discussion.
Line 287-316: A likely alternative explanation is that fish in familiar groups are more at ease as they are with individuals they have seen before.
Line 342-343: Some more sentences about the relevance of studying such a fast and innate response as studied here would be helpful as it may explain the lack of significant results.
Line 354-355: I wouldn’t call one significant effect far greater
Line 361: The test used was not novel altought the sentence currently seems to suggest that.
Line 363-366: Please see other comments about the underlying analysis and be careful about statements about behaviour being ‘adaptive’ as finding significant relationships between behaviours does not require any adativeness.
Comment 7 Reviewer 2: This reviewer makes a fair point that I don’t think the authors address satisfactory. The authors should at least make the suggestion by the reviewer clearer in their discussion. With the current analyses the authors did (i.e. not controlling for group) it is hard to know for certain if it was indeed not always individuals of a certain size responding quicker.

---

## Round 0.3 · Minor Revisions

· Academic Editor

Minor Revisions

Your previous submission was reviewed by two reviewers (Reviewer 3 and Reviewer 4). Unfortunately, Reviewer 4 was not available to review your latest submission, but Reviewer 3 has. Additionally, I have thoroughly reviewed both your revised submission as well as your responses to Reviewer 4's comments, which I believe you have answered satisfactorily.

At this stage, the comments provided by myself and Reviewer 3 are, for the most part, editorial in nature (mostly suggestions to help clarify your language). I believe that if you are able to address each of the very minor comments and suggested edits provided by myself and Reviewer 3 I will be able to accept your article for publication in PeerJ.

My suggested edits are as follows:
Line 80, please remove the space after the bracket and before the comma i.e. “…mobbing (Grabowska-Zhang et al. 2012) , or perform predator inspection (Dugatkin & Godin…” should be “…mobbing (Grabowska-Zhang et al. 2012), or perform predator inspection (Dugatkin & Godin…”
Line 85, I think you should use the past tense i.e. “For example, studies to date have focused exclusively on the effect of familiarity on the latency of the response (Griffiths et al. 2004; Strodl & Schausberger 2012) and have not considered other aspects of the escape performance.”
Line 89, please add commas here to break up the subclause i.e. “For instance, latency, considered as the time between the onset of the predator attack and the start of the response, is crucial for the outcome of the interaction (Fuiman et al. 2006).”
Line 103, please delete “Paolo”
Methods: how old (approximately) were the fish that you tested?
Line 149, please add a comma after “further” i.e. “Further, in all stock tanks there are large and smaller boulders and java moss...”
Line 151, when you state “Juveniles were allocated to a holding tank (20 x 22 x 30 cm) to create a test group”, please provide the total number of juveniles put into each test group.
Line 161 (and also line 227) should read “…composed of three juveniles…” rather than “composed by three juveniles”
Line 162, I think this sentence would be more clear if written thus: “In familiar groups, individuals were tested with those fish that they shared the holding tank with for two weeks prior to testing.”
Line 164, I think this sentence would be more clear if written thus: “For unfamiliar groups, we took three fish, each from a different holding tank so they had not seen each other before, and put them together in the observation chamber for testing (Figure 1).”
For your data analysis, who conducted the video coding? Was inter-rater reliability performed with a subset of the videos? If so, please report that procedure and the concordance of raters’ cording.
Line 265, please delete “however”
Line 294, if I have interpreted your meaning correctly, I think rather than writing “In both studies…” it would be clearer to say “In both their and our study…”
Line 196, please add commas before and after “therefore” i.e. “…results, therefore, suggest…”

Reviewer 3 ·

Basic reporting

no comment

Experimental design

no comment

Validity of the findings

no comment

Additional comments

Comments on Wolcott, Barbosa and Ojanguren 9-11-17
I have studied the responses to Reviewer 1 & 2, and generally I’m satisfied that you have adequately addressed their concerns, as well as my own (#3). I believe that statistical analysis is much improved over the original, and I agree with the decision not to use an analysis of residuals.
I think you have gotten an interesting result in the finding that familiarity does not affect all components of the escape response. The findings relative to size effects are also noteworthy. I think these will be useful to those working in this field. One of my thoughts on the results is that individuals in groups of familiar individuals ‘feel safer’ and hence react less to stimuli than those among unfamiliar individuals. I might avoid saying that they feel more at ease, which, unlike ‘feel safer’, is not standard in the behavioral literature, as far as know.
Below. I have made more suggestions. Most are of an editorial nature. I think the abstract was generally overlooked in the original reviews, because it seems to need a bit more work.
24 I would take out “antipredator”. All you need for clarity is “escape responses”
24 Suggest “the guppy, Poecilia reticulata”. Drop “Trinidadian”
30 ‘surprisingly’ = ‘contrary to the prediction’
30 ‘by’ instead of ‘of’ ?
31-32 Take out the plausible sentence, as this opinion is better left for the discussion
33 insert ‘rather’ after ‘size’
35-36. I think you should revise this. Rather than advocating for your ‘approach’, which is not the subject of this study, try a sentence that summarizes or gives your most salient conclusion. How about the last sentence of the Discussion? Something that grabs the reader’s attention.
64 insert ‘and’ after the comma
65 ‘thus’ take out
109-112 Somehow, I think this sentence is out of place. It doesn’t logically lead to 112, as yours is not an integrated approach, right?
113 ‘shaping’ = ‘affecting’
115-116 Here’s a suggested rewording. Guppies shoal . . . . birth. During this early developmental period, guppies acquire group familiarity and its associated antipredator benefits ( ). Does such familiarity then lead to greater group cohesion, e.g. as in avoid joining a different group?
However, the whole paragraph seems not to hang together very well. The points are not flowing logically to me. Suggest revisiting it. I think you need to better set the stage for you important last paragraph, the statement of purpose.
120 doesn’t flow logically from 119.
170 ‘by’ instead of ‘of’
172-175. Here’s another suggestion for wording. ‘Familiar groups consisted of individuals that were together for two weeks. Unfamiliar groups were comprised of individuals taken singly from different holding tanks.’
You can omit the rest, i.e. …not seen each other before and put them together…
232 typo, check ‘…both response variables,
231-234 The ANOVA model assumes normality and homogeneity of variances of residuals, and also independence of residuals when you analyze responses on the dependent variable. I think you might simply say that normality and homogeneity assumptions about the distributions of residual values on the dependent variable were improved by log-transforming the response variables. The analysis is generally robust to small deviations, and it is usually pretty hard to absolutely meet all the assumptions. I think your results are clear enough. That’s my opinion.
239 Try ‘Individual S.L. did not differ…
240 (SD)
278 ‘quality’. This word seems a little vague to me. Not sure exactly what it means. You might say ‘… some aspects of the escape response’ or something to that effect.
298. Why not ‘feel safer’ rather than ‘at ease’?
367 Again, maybe ‘quality’ should be ‘specific components of’

I trust that you will find these comments useful overall.
Doug Fraser

---

## Round 0.4 · accepted · Accept

· Academic Editor

Accept

Thank you very much for so quickly and carefully responding to each of my and Reviewer 3's comments and suggested edits. I am happy to accept your article for publication in PeerJ.